# Network Security Prediction of Industrial Control Based on Projection Equalization Optimization Algorithm

**DOI:** 10.3390/s24144716

**Published:** 2024-07-20

**Authors:** Guoxing Li, Yuhe Wang, Shiming Li, Chao Yang, Qingqing Yang, Yanbin Yuan

**Affiliations:** 1School of Computer Science and Information Engineering, Harbin Normal University, Harbin 150025, China; liguoxing@stu.hrbnu.edu.cn (G.L.); shimingli@hrbnu.edu.cn (S.L.); yangqq@stu.hrbnu.edu.cn (Q.Y.); yuanyanbin@stu.hrbnu.edu.cn (Y.Y.); 2School of Physics and Electronic Engineering, Harbin Normal University, Harbin 150025, China; yc_hit@hrbnu.edu.cn

**Keywords:** belief rule base (BRB), evidence reasoning (ER), industrial control systems (ICSs), projection equalization optimization algorithm (P-EO), security situation prediction

## Abstract

This paper predicts the network security posture of an ICS, focusing on the reliability of Industrial Control Systems (ICSs). Evidence reasoning (ER) and belief rule base (BRB) techniques are employed to establish an ICS network security posture prediction model, ensuring the secure operation and prediction of the ICS. This model first integrates various information from the ICS to determine its network security posture value. Subsequently, through ER iteration, information fusion occurs and serves as an input for the BRB prediction model, which necessitates initial parameter setting by relevant experts. External factors may influence the experts’ predictions; therefore, this paper proposes the Projection Equalization Optimization (P-EO) algorithm. This optimization algorithm updates the initial parameters to enhance the prediction of the ICS network security posture through the model. Finally, industrial datasets are used as experimental data to improve the credibility of the prediction experiments and validate the model’s predictive performance in the ICS. Compared with other methods, this paper’s prediction model demonstrates a superior prediction accuracy. By further comparing with other algorithms, this paper has a certain advantage when using less historical data to make predictions.

## 1. Introduction

Industrial Control Systems (ICSs) are integrated and critical control systems consisting of both hardware and software components interconnected via networks to support the operation and security of critical infrastructure. Their applications are diverse, with the majority being utilized in facilities such as power plants, sewage treatment plants, and other critical infrastructure factories [1]. Technological advancements and the expansion of factories have led to many ICSs needing to interact with external networks. However, due to inadequate security defenses in some ICSs, they are susceptible to novel cyberattacks during network interactions, which can disrupt the normal operation of companies [2]. Therefore, it is crucial to prioritize security predictions for ICSs.

The prediction of an Industrial Control System’s (ICS) network security posture is an effective means of preventing network security incidents [3]. The predicted results can prevent network attacks and provide a basis for administrators to take necessary measures in advance. However, ICSs feature characteristics such as diversity, heterogeneity, and high security, necessitating the fulfillment of high robustness and security requirements [4]. Moreover, the complex and changeable production environment of ICSs, interference-prone data extraction, and the large data volume present challenges in data acquisition [5,6]. Inherent uncertainty in the data obtained from ICSs, which includes both probabilistic and fuzzy uncertainties, complicates the establishment of predictive models for ICS network security [7]. Therefore, establishing predictive models requires a comprehensive analysis of additional information to efficiently forecast the network security posture of ICSs. Additionally, understanding popular forecasting methods in related fields is also essential.

In recent years, there has been a growing number of various network attack incidents, resulting in significant economic losses and disruptions to daily life. This has led many experts to increasingly emphasize the direction of security prediction. Nowadays, numerous different security prediction methods have been proposed. Yin et al. proposed a situational prediction model combining a Time Series Convolutional Network (TCN) with a Transformer [8]. Similarly, Xu et al. introduced an intelligent prediction algorithm to address real-time security performance issues and improve Convolutional Neural Network (CNN) models [9]. Sepasgozar et al. utilized federated learning and Long Short-Term Memory (LSTM) algorithms for network traffic prediction [10]. Li et al. employed a convolutional neural network approach for privacy protection prediction [11]. Qi et al. focused on privacy-aware data in urban industrial environments, combining local sensitive hash techniques for data fusion and prediction [12]. Liu et al. addressed short-term wind power prediction using discrete wavelet transformation and LSTM technology [13]. Riihijarvi et al. applied machine learning techniques to wireless network performance prediction problems [14]. Wen et al. proposed a semi-supervised prediction model for solving prediction function problems [15].

Based on the prediction methods and information used in the modeling above, security prediction can be classified into three methods.

(1) Qualitative knowledge-based methods: These methods rely on experts’ practical experience and various factors to determine the weights of model factors and appropriate algorithms for prediction. Examples of studies using them include that by Ma et al., who proposed an effective online prediction algorithm for Petri net marking prediction [16]. Tehseen et al. proposed an algorithm for earthquake prediction using expert systems [17]. Xi et al. used the analytic hierarchy process to establish evaluation indicators for solid mineral exploration and target areas [18]. However, solely relying on qualitative knowledge-based methods may not be sufficient to establish accurate prediction models due to the complexity of ICS network structures and the suddenness of virus attacks. Moreover, this method is primarily based on expert experience, which is subjective and may lead to significant errors if expert experience is inadequate. Additionally, various types of uncertain information cannot be effectively utilized.

(2) Quantitative data-based prediction methods: These methods involve the use of artificial intelligence to establish relevant mathematical models, which are then trained using a large amount of related data. Examples of studies using them include that by Ge et al., who utilized a high-dimensional Bayesian regression framework and multi-gene risk scores to address the multi-gene prediction problem [19]. Wang et al. employed traditional linear regression, Factsage calculations, and backpropagation (BP) neural networks to predict the deformation temperature of coal ash [20]. Jin et al. designed a plane flow variational auto-encoder prediction model (PFVAE) using time series methods [21]. Liu et al. proposed an end-to-end deep learning architecture for predicting subway entrance and exit passenger flow [22]. Chen et al. tackled the prediction accuracy issue in network situational awareness by proposing a generalized radial basis function [23]. Speiser et al. were able to assess different variable selection techniques in random forest classification settings [24]. This method operates as a black box, and its operational mechanism cannot provide reasonable explanations. Therefore, it is challenging to apply this method in complex ICS network security situation prediction. Furthermore, obtaining good model parameters is difficult for small-scale samples using quantitative data-based prediction methods, leading to a reduced prediction accuracy.

(3) Semi-quantitative information-based prediction methods: These methods integrate qualitative knowledge and quantitative data. They use qualitative knowledge to determine the parameters of the prediction model and train the prediction model using a large amount of quantitative data for prediction. Some examples of studies using them include that by Dong et al., who applied the extended Markov model to evaluate and predict the status of spiral motors [25]. Cao et al. proposed several improved fuzzy rough neural network models and validated their advantages through experiments on complex stock time series prediction tasks [26]. Zhang et al. used Bayesian and automated machine learning methods to tackle the performance prediction problem of tunnel boring machines [27]. These methods combine qualitative knowledge and quantitative data, using expert experiences to establish preliminary models, ensuring the accurate prediction of complex ICS network security posture situations with fewer data samples. However, existing semi-quantitative information-based methods can only handle single types of uncertain information, and expert knowledge may also be affected by external and internal factors, thereby influencing the prediction results.

From the above methods, it can be seen that the first two approaches consider only single knowledge or data sources. The third approach, while combining and improving upon the first two methods, can handle only single types of data and overlooks the uncertainty of expert knowledge. Several scholars have proposed solutions as well. For example, Wang et al. proposed the ER algorithm [28] and the BRB method [29], introducing a novel logical data processing approach. Yang et al. introduced a new model integrating the extended belief rule-based system (EBRBS) and evidence reasoning rules for environmental investment prediction [30]. Cheng et al. applied the BRB to fault detection in flywheel systems, proposing a Fault Diagnosis Model (FFBRB) based on fuzzy fault tree analysis theory [31]. Li et al. studied complex systems and embedded expert knowledge into transformation matrices based on rule change techniques, proposing a health assessment model based on ER rules [32]. Zhang et al. addressed the problem of setting parameters rationally due to the increase in antecedent attributes, proposing a method to automatically generate large-scale BRB initial parameters using partial standard rules and cloud models [33]. He et al. applied ER and the BRB to the field of wireless sensor network fault prediction, proposing a wireless sensor fault prediction method [34]. Han et al. proposed a model parameter optimization method using interval optimization strategies to predict lithium battery capacity, also employing the Whale Optimization Algorithm (WOA) [35]. Hu et al. introduced a new network security situation prediction model using Hidden BRB models and the Covariance Matrix Adaptation Evolution Strategy (CMA-ES) algorithm [36].

Through the analysis above, this study applies ER and the BRB to construct a predictive model. When predicting the security situation of heterogeneous networks in ICSs, it is essential to consider the actual situation. Due to the complexity of industrial environments, heterogeneous networks are even more intricate. Therefore, it is intended to select the best parameters of the prediction model through the EO algorithm [37]. Due to the characteristics of industrial control heterogeneous networks, projection operation is added based on the EO algorithm. Then, the P-EO algorithm is used to mitigate its impact and enhance the prediction accuracy. Since semi-quantitative information is also crucial, it requires careful processing. However, when dealing with excessive data, it may lead to the challenge of BRB combination explosion. To tackle this issue, this study employs the ER iteration algorithm to integrate semi-quantitative information. The BBR model offers a visually intuitive reasoning process and a rigorous structure. It effectively addresses the challenge of poor modeling effectiveness in industrial control heterogeneous networks, attributed to their large scale and data deficiencies [38]. This enables managers to obtain more reliable prediction results and accurate information, thereby reducing company losses and avoiding risks. Consequently, it enhances the risk resistance and emergency response capabilities of industrial control heterogeneous network systems.

The structure of this paper is as follows: The second section describes the problem of predicting the security posture of ICS networks and provides an overview of the process. The third section introduces the construction of the ICS network security situational prediction model, providing further details on its development process. The fourth section involves testing the predictive model using specific case data and comparing this with other methods to evaluate its practicality. Finally, the fifth section summarizes the findings and presents the outlook for this research field.

## 2. Problem Description

This section will be divided into three parts to introduce the prediction of ICS network security:

(1) Given the complexity of ICSs, conducting network security posture assessment is essential. Evaluation indicators are derived from impact analysis, and subsequently, an evaluation framework is established based on their significance. The multitude and diversity of evaluation indicators make data fusion challenging, hence the adoption of the ER iteration algorithm to mitigate this issue.

(2) A network security posture prediction model is established based on the BRB, where the network security posture values of adjacent time periods are used as inputs to predict the network security posture of the next time period.

(3) Expert knowledge in setting initial parameters may lead to significant errors in the prediction model. Therefore, the P-EO algorithm is adopted to update the initial parameters of the BRB model to address this issue.

### 2.1. Parameter Table

All parameter descriptions are summarized in Table 1.

### 2.2. Industrial Control Network

The industrial control network exhibits heterogeneity and can be classified based on deterministic latency. Communication in heterogeneous networks can be categorized into integrated and interconnected modes. The integrated mode is suitable for scenarios with weak latency requirements, while the interconnected mode is suitable for scenarios with stronger latency requirements. A structural diagram is constructed based on the characteristics of heterogeneous networks and the IEC 62264-1 standard [39], illustrating the specific architecture of ICSs, as depicted in Figure 1.

By observing the structure of ICSs, it can be divided into five layers from bottom to top according to different functions:

(1) Field device layer: this layer includes various types of sensing devices and actuator units used for perceiving and operating the production process.

(2) Field control layer: this layer includes various types of controller units used for controlling the actuator devices.

(3) Process monitoring layer: this layer includes monitoring servers used for managing the production process.

(4) Production management layer: this layer includes PLMS and management servers used for managing the production process.

(5) Enterprise management layer: This layer includes functional units such as web servers, which provide decision-making capabilities for employees at the enterprise decision-making level.

To better facilitate prediction and data integration, this paper divides ICSs into information networks and control networks. Regarding the information network component, which encompasses the enterprise management layer, it possesses the capability to quickly integrate and process data from lower layers and requires extensive interaction with external networks for data processing. As for the control network component, this encompasses the production management layer, process monitoring layer, field control layer, and field device layer. Situated at the lowest layer, it is more vulnerable to attacks. Continuously acquiring information is essential for promptly informing administrators and enabling them to take emergency measures in case of an attack.

Due to the structural characteristics of ICSs, they are susceptible to various network attacks. Many ICSs are not directly connected to external networks; thus, enabling network access poses a risk of system vulnerability to network viruses, potentially causing damage to the company. Generally, ICSs establish complete architectures and protocols during their manufacture. However, over time, these architectures and protocols may become vulnerable to new attack methods, revealing vulnerabilities and flaws.

### 2.3. Fusion of Evaluation Indicators

After dividing the ICS structure, it is necessary to analyze how to integrate its evaluation indicators. To address the network security posture of ICSs and integrate evaluation indicators more effectively, this paper categorizes the evaluation indicators into four levels to manage the relevant data and establish an ICS assessment model. Initially, data from the fourth-level framework undergo fusion using ER, combining the attack frequency and attack severity of the evaluation indicators to derive results for the third-level evaluation indicators. Secondly, ER fuses the results of the third-level evaluation index to obtain the second-level evaluation index results. Subsequently, ER further integrates the results of the second-level evaluation indicators, ultimately yielding the network security situation assessment results. The model is formulated as Equation (1).
(1)$$\left\{\begin{array}{c} A_{i}=ER\left(a_{i},q_{i}\right)\;\left(i=1,2,\cdots ,n;\;x=1,2,\cdots ,n_{1}\right)\\ A_{j}=ER\left(a_{j},q_{j}\right)\left(\;j=1,2,\cdots ,n_{2}\right)\\ B_{z}=ER\left(A_{i},A_{j}\right)\left(\;Z=1,2\right)\\ C=ER\left(B_{z}\right) \end{array}\right.$$$a_{i}$ represents the frequency of various attacks in the i-th evaluation indicator and $q_{i}$ represents the severity of the corresponding attack in the i-th evaluation indicator. $A_{i}$ represents the fused result of the i-th third-level evaluation indicator. $B_{z}$ represents the second-level evaluation indicators obtained through the fusion of the third-level evaluation results. *C* represents the first-level evaluation indicators, derived by integrating the second-level evaluation results. This serves as the comprehensive fused assessment result of the ICS network security posture. ER() denotes the process of merging the evaluation indicator data based on *ER* iteration.

### 2.4. Network Security Posture Prediction and Model Optimization

Once the evaluation indicators are fused, the network security posture assessment result is obtained. When this is achieved along with the construction of the initial parameters, it is appropriate to build the predictive model using the *BRB*. The construction of the model is formulated as Equation (2).
(2)$$y=BRB\left(O\left(k-1\right),O\left(k\right),\partial \right)$$

Here, y represents the model’s prediction result. BRB() denotes the nonlinear process of deriving the prediction result using BRB technology. $O\left(k-1\right)$ denotes the network security posture value at time *k* − 1, while $O\left(k\right)$ denotes the network security posture value at time *k*, and ∂ represents the parameter set of the BRB, which is determined by experts.

During prediction, the construction of expert parameters may not always be reliable. Factors such as the actual network situation and network equipment can influence expert knowledge. To mitigate this influence, this paper proposes the P-EO algorithm for parameter optimization. Through this algorithm, model parameters are optimized to enhance the prediction accuracy and achieve satisfactory prediction results.

## 3. Industrial Control System Network Security Posture Prediction Model Based on P-EO

### 3.1. Prediction Process

This article divides the network security posture prediction process into three steps, as illustrated in Figure 2:

Step 1: Based on factors such as the structure of the ICS, select representative evaluation indicators, and establish the actual network security posture assessment framework.

Step 2: Utilize the evaluation framework established for the ICS, apply ER fusion to integrate data from various layers, and derive their network security posture values.

Step 3: Develop a predictive model using the BRB and optimize parameters using the P-EO algorithm to minimize prediction errors.

### 3.2. Establishment of Evaluation Framework

When conducting the model predictions in this paper, the first step is to establish an evaluation framework. This paper considers the structure, security aspects, and types of attacks in the ICS, selecting representative indicators as evaluation metrics. The resulting four-level evaluation framework is detailed in Table 2.

Establishing the evaluation framework enables the better analysis and organization of data. According to Table 2, it can be observed that both the information network and the control network are categorized as first-level indicators. Considering that different devices may face various threats of network attacks, the information network is prone to attacks due to frequent information transmission, exchange, and sharing. Therefore, the second-level indicators of the information network are different types of network attacks. The control network includes various information and network devices, such as sensors and switches, which are crucial for system operation. The second-level indicators of the control network are significant devices, as network attacks may target these devices. Therefore, the third-level indicators of the control network are the types of network attacks corresponding to its devices. Finally, the attack frequency and severity are used as the ultimate evaluation metrics. The attack frequency is determined by the number of attacks of each type within a unit time period. The severity of attacks is determined based on standards set by relevant experts.

Following the establishment of the evaluation framework, the model proceeds to determine the weights of the evaluation indicators. Based on the importance of each layer of evaluation indicators, weights (ω) are assigned to the evaluation indicators (r). In ICSs, the impact of an evaluation indicator on the assessment result increases with its data variability; thus, indicators with larger data variations are assigned higher weights. The coefficient of variation method is employed to effectively determine these weights. The specific process is outlined as follows:

Step 1: Initial Matrix

Generate the initial matrix Y using evaluation data.
(3)$$Y={\left(y_{ij}\right)}_{m\times n},i=1,2,\cdots ,m;j=1,2,\cdots ,n$$
where $y_{ij}$ represents the *j*th evaluation value in the ith sample, *m* represents the maximum number of samples contained, and n represents the maximum number of evaluation indicators.

Step 2: Standardization

Each indicator may have different magnitudes, so it is necessary to scale them to the same range.
(4)$$y_{ij}=\frac{y_{ij}-min\left(y_{ij},\cdots ,y_{nj}\right)}{max\left(y_{ij},\cdots ,y_{nj}\right)-min\left(y_{ij},\cdots ,y_{nj}\right)}$$

Step 3: Mean Calculation

Calculate the mean $A_{j}$ for each assessment indicator.
(5)$$A_{j}=\frac{1}{n}\sum_{i=1}^{g}y_{ij}$$

Step 4: Standard Deviation Calculation

Calculate the standard deviation $w_{j}$ for each assessment indicator.
(6)$$S_{j}=\sqrt{\frac{1}{n}\sum_{i=1}^{g}{{(y}_{ij}-A_{j})}^{2}}$$

Step 5: Coefficient of Variation Calculation

Calculate the coefficient of variation for each assessment indicator.
(7)$$V_{j}=\frac{S_{j}}{A_{j}}$$

Step 6: Weight Calculation

Calculate the weight $w_{j}$ for each assessment indicator.
(8)$$w_{j}=\frac{V_{j}}{\sum_{j=1}^{x}V_{i}}$$

### 3.3. Network Security Posture Assessment Based on ER

After establishing the evaluation indicators, the next step involves utilizing the indicator data to assess the network security posture. Each data point holds unique significance and contributes to the final evaluation result. This paper employs ER iteration to progressively integrate relevant indicator data, thereby obtaining the fused result. The process unfolds as follows:

Step 1: Initialization

Assume a set of basic attributes $\left\{r_{1},r_{2},\cdots ,r_{j},\cdots {,r}_{N}\right\}$ constitutes the evaluation system, with corresponding weights $\left\{\omega_{1},\omega_{2},\cdots ,\omega_{j},\cdots ,\omega_{N}\right\}$, and 0 ≤$\;\omega_{i}\;$≤ 1. The evaluation level is denoted as P, with N levels. The description of the evaluation indicators for each level is as follows:(9)$$r_{j}=\left(P_{n},\alpha_{n,j}\right),\left(\Theta ,\alpha_{\Theta ,j}\right),j=1,\cdots ,L;n=1,\cdots N$$

Step 2: Basic Probability Mass

The corresponding basic probability mass is calculated using the confidence level ${\;\alpha }_{i,j}$, as follows:(10)$$U_{n,j}=\omega_{j}{\;\alpha }_{n,j}$$
(11)$$U_{\vartheta ,j}={1-\omega }_{j}\sum_{j=1}^{M}{\;\alpha }_{n,j}$$
(12)$${\bar{U}}_{\vartheta ,j}=1-\omega_{j}$$
(13)$${\tilde{U}}_{\vartheta ,j}=\omega_{j}\left(1-\sum_{n=1}^{N}{\;\alpha }_{n,j}\right)$$

Step 3: ER Iterative

(a) Combinatorial fundamental probability quality

The combined probability mass is obtained through the basic probability masses. The formula is as follows:(14)$$U_{n,1}={\bar{U}}_{\vartheta ,1}+{\tilde{U}}_{\vartheta ,1}$$
(15)$$U_{n,r\left(2\right)}=F_{0}\left[U_{n,1}U_{n,2}+U_{n,1}U_{\vartheta ,2}+U_{\vartheta ,1}U_{n,2}\right]$$
(16)$${\tilde{U}}_{\vartheta ,r\left(2\right)}=F_{0}\left[{\tilde{U}}_{\vartheta ,1}{\tilde{U}}_{\vartheta ,2}+{\tilde{U}}_{\vartheta ,1}{\bar{U}}_{\vartheta ,2}+{{\bar{U}}_{\vartheta ,1}\tilde{U}}_{\vartheta ,2}\right]$$
(17)$${\bar{U}}_{\vartheta ,r\left(2\right)}=F_{0}{\bar{U}}_{\vartheta ,1}{\bar{U}}_{\vartheta ,2}$$
(18)$$F_{0}={\left[1-\sum_{i=1}^{N}\sum_{j=1,i\neq j}^{N}U_{i,1}U_{j,2}\right]}^{-1}$$

(b) Combining Confidence

The formula for combining confidence levels is as follows:(19)$$\alpha_{n,r\left(2\right)}=\frac{U_{n,r\left(2\right)}}{1-{\bar{U}}_{\vartheta ,r\left(2\right)}},i=1,\ldots ,N$$
(20)$$\alpha_{\vartheta ,r\left(2\right)}=\frac{{\tilde{U}}_{\vartheta ,r\left(2\right)}}{1-{\bar{U}}_{\vartheta ,r\left(2\right)}}$$
(21)$$r\left(2\right)=\left(P_{n},\alpha_{n,r\left(2\right)}\right),\left(\Theta ,\alpha_{\Theta ,e\left(2\right)}\right),n=1,\cdots N$$

(c) Final Confidence

The synthesized basic probability masses are combined with the subsequent evidence in a loop, alternating between steps (a) and (b), ultimately calculating the final result. The formula is as follows:(22)$$r\left(L\right)=\left(P_{n},\alpha_{n,r\left(L\right)}\right),\left(\Theta ,\alpha_{\Theta ,e\left(L\right)}\right),n=1,\cdots N$$

(d) Fusion Result

The expected utility of the evaluation, assuming the utility of evaluation level $P_{n}$ is $u(P_{n})$, is as follows:(23)$$u=\sum_{n=1}^{N}u\left(P_{n}\right)\alpha_{n,r\left(L\right)}$$

The fusion result will be constrained between 0 and 1, where smaller values indicate a safer state.

### 3.4. Network Security Posture Prediction Based on BRB

Once the network security posture assessment results of the system are obtained, the next step is to proceed with the prediction work. This paper integrates adjacent time series to obtain the network security posture value for the next moment. The BRB model integrates the values at time *k* − 1 and time kto obtain the network security posture value for the next moment. The process is detailed as follows:(24)$$R_{k}:If\;T\left(k-1\right)\;is{\;B}_{1}^{k}\wedge T\left(k\right)\;is{\;B}_{2}^{k}\;Then\;T\left(k+1\right)\;is\;\left(D_{1},\beta_{1,k}\right),\cdots ,\left(D_{N},\beta_{N,k}\right)\;With\;rule\;weight{\;\theta }_{k}\;and\;attribute\;weight{\;\delta }_{1},\delta_{2}$$

Here, $R_{k}$ represents the kth belief rule, ${\;B}_{1}^{k}$ and $B_{2}^{k}$ represent the reference values corresponding to the two premise attributes of the *k*th rule. $D_{1}\cdots ,D_{N}$ represent N results, and $\beta_{1,k}\cdots ,\beta_{N,k}$ are the confidence levels associated with all N results in the *k*th belief rule. $\theta_{k}$ represents the weight of the *k*th belief rule, while $\delta_{1}$ and $\delta_{2}$ denote the weights assigned to two antecedent attributes.

When performing network security posture prediction, the model derivation requires ER analysis for deduction. The specific steps are as follows:

Step 1: Attribute Matching

The matching degree between the input sample information and the confidence rules needs to be calculated as follows:(25)$$\alpha_{k}^{i}=\left\{\begin{array}{c} \frac{W_{k}^{l+1}-V_{k}^{*}}{W_{k}^{l+1}-W_{k}^{l}},i=l,W_{i}^{l}\leq V_{k}^{*}\leq W_{k}^{l+1}\\ 1-\alpha_{k}^{i},i=l+1\\ 0,i=1\cdots I,k\neq l,l+1 \end{array}\right.$$

Step 2: Activation Weight Calculation

Once the rule is successfully matched, the corresponding rule will be activated, and its activation weight is calculated as follows:(26)$$w_{k}=\frac{\theta_{k}\prod_{k=1}^{M}{\left(\alpha_{k}^{i}\right)}^{{\;\delta }_{i}}}{\sum_{l=1}^{K}\theta_{k}\prod_{i=1}^{M}{\left(\alpha_{k}^{l}\right)}^{{\;\delta }_{i}}}$$

Step 3: ER Analysis

After calculating the activation weights, the activated rules need to be combined. This is achieved through ER inference for rule synthesis, calculated as follows:(27)$$\beta_{n}=\frac{\mu \left[\prod_{l=1}^{L}\left(w_{l}\beta_{n,l}+1-w_{l}\sum_{i=1}^{N}\beta_{i,l}\right)-\prod_{l=1}^{L}\left(1-w_{l}\sum_{i=1}^{N}\beta_{i,l}\right)\right]}{1-\mu \left[\prod_{l=1}^{L}1-w_{l}\right]}$$
(28)$$\mu =\frac{1}{\begin{array}{c} \sum_{n=l}^{N}\prod_{l=1}^{L}\left(w_{l}\beta_{n,l}+1-w_{l}\sum_{i=1}^{N}\beta_{i.l}\right)-\\ \left(N-1\right)\prod_{l=1}^{L}\left(1-w_{l}\sum_{i=1}^{N}\beta_{i,l}\right) \end{array}}$$

Step 4: Utility Calculation

After computing the confidence levels for each assessment grade, the prediction is obtained through a utility calculation, as follows:(29)$$T\left(k+1\right)=\sum_{n=1}^{N}u\left(D_{n}\right)\beta_{n}$$

### 3.5. Optimization of BRB Model Based on P-EO Algorithm

To address the uncertainty associated with expert knowledge in setting the initial parameters, this study employs a P-EO algorithm for model optimization. By leveraging projection to manage the constraints of the BRB, the P-EO algorithm enhances the BRB model, thereby improving its predictive accuracy.

The optimization and constraint description of the prediction model is outlined as follows:(30)$$min\;MSE\left(\theta_{k},\beta_{n,k},\delta_{i}\right)\;s.t.\;\;0\leq \beta_{n,k}\leq 1,n=1,\cdots ,N,k=1,\cdots ,K\;0\leq \theta_{k}\leq 1,k=1,\cdots ,K\;0\leq \delta_{i}\leq 1,i=1,\cdots ,M\;\sum_{n=1}^{N}\beta_{n,k}=1,k=1,\cdots ,K$$

The model updates the initial parameters set by experts through optimization algorithms, thereby enhancing the predictive performance of the model. The symbol *MSE* represents the mean squared error between the predicted network security posture values of the forecasting model and the actual network security posture values, determining whether the model accurately predicts the security situation. The formula is outlined as follows:(31)$$MSE\left(\theta_{k},\beta_{n,k},\delta_{i}\right)=\;\frac{1}{T}\sum_{t=1}^{T}{\left({output}_{estimated}-{output}_{actual}\right)}^{2}$$
where ${output}_{estimated}$ represents the actual network security posture value of the ICS, and ${output}_{actual}$ represents the predicted network security posture value of the ICS. The formula is ${output}_{estimated}=\sum_{n=1}^{N}u\left(D_{n}\right)\beta_{n}$. Here, T represents the number of samples used for training. This paper employs the P-EO algorithm to reduce the mean squared error of the model. A lower mean squared error indicates a closer approximation to the actual network security posture, thus improving the accuracy. The computational process of the P-EO optimization algorithm is depicted in Figure 3, and the specific calculation process is as follows:

Step 1: Initialization

Initialize vector $C^{0}$ as the initial expectation of the P-EO algorithm.
(32)$$C^{0}=\{\theta_{1}\cdots \theta_{k},\beta_{1,1}\cdots \beta_{N,K},\delta_{1}\cdots \delta_{M},ub,lb\}$$
(33)$$C_{i}^{0}=C_{min}+r_{i}\left(C_{max}-C_{min}\right),i=1,2,\cdots ,n$$

Step 2: Projection and Adaptive Values

Due to the limitations of the EO algorithm on the constraints of industrial control heterogeneous networks, that is, some candidate solutions do not meet the constraints, but they conform to the actual situation, the mapping is carried out by projection, and then the candidate solutions meet the conditions. The EO algorithm is made more effective in making predictions. After projection, the adaptive value needs to be calculated, that is, whether the updated parameters can achieve a good prediction result. The mean squared error (*MSE*) serves as the objective function, while $C_{eq,k}$ represents the parameters of the inference process.
(34)$$C_{k}^{g+1}\left(1+\nu n_{e}\times \left(xn-1\right):\nu n_{e}\times xn\right)\;\;=C_{k}^{g+1}\left(1+\nu n_{e}\times \left(xn-1\right):\nu n_{e}\times xn\right)\;-A_{e}^{T}\times {\left(A_{e}\times A_{e}^{T}\right)}^{-1}\times C_{k}^{g+1}\left(1+\nu n_{e}\times \left(xn-1\right):\nu n_{e}\times xn\right)\times A_{e}$$
(35)$$C_{eq,k}=min\;MSE\left(C_{k}^{g+1}=\{(\theta ,\beta ,\delta ,ub,lb)\;\}\right)\;s.t.\;0\leq \delta \leq 1;0\leq \beta \leq 1;0\leq \delta \leq 1;ub=1,lb=0$$

Step 3: Equilibrium State Pools

To enhance its global optimization capability and obtain better local optimal solutions, five current optimal solutions are selected from the samples. After selecting the balanced state (see Figure 4), the candidate solutions’ balance pool is as follows:(36)$$C_{eq,pool}={\{C}_{eq,I},C_{eq,II},C_{eq,III},C_{eq,IV},C_{eq,ave}\}$$

Step 4: Update Exponential Coefficients

To facilitate both local and global searches more effectively, exponential coefficients are introduced, and are computed as follows:(37)$$F=a_{1}\times sign\left(r-0.5\right)\left[e^{-\lambda_{l}t}-1\right]$$

Step 5: Update Quality Generation Coefficients

To better explore local optimal solutions, the generation rate is restricted. The calculation is as follows:(38)$$G=G_{CP}\left(C_{eq}-\lambda_{l}C\right)$$
(39)$$G_{CP}=\left\{\begin{array}{c} 0.5r_{i},if\;r_{2}\geq 0.5\\ 0,otherwise \end{array}\right.$$

Step 6: Update Individual Current Solution

For the optimization problem, the individual solution is updated as follows:(40)$$C_{k}^{g+1}=C_{eq}+\left(C_{k}^{g+1}-C_{eq}\right)F+G\left(1-F\right)/\lambda_{l}V$$

Repeat steps two to six until the iteration count is met. When the iteration count is reached, the loop will terminate, yielding the optimal parameters.

## 4. Case Study

This section aims to verify the predictive capability of the model through experiments. We utilize the X-IIoTID Dataset [40] and TON-IoT [41,42,43] dataset as experimental data to evaluate the model’s performance indicators at the first level. These datasets provide real-time status information for the ICS. Based on these data, we establish a prediction model and compare its effectiveness with other methods.

### 4.1. Problem Statement

The X-IIoTID Dataset used is an intrusion dataset that is independent of connections and devices, encompassing multiple attack types and protocols. It serves as a dataset for control networks, focusing on attacks targeting historical/real-time databases, asset management systems, and industrial gateways. Historical/real-time databases are primarily subjected to three types of attacks: vulnerability scanning, general scanning, and erroneous data injection. Asset management systems face attacks such as ransomware, ransom denial-of-service, and discovery-assisted attacks. Industrial gateways encounter attacks like Modbus register reading, brute force attacks, and reverse shell attacks. The TON-IoT dataset used contains heterogeneous data sources such as IoT service remote sensing datasets and network traffic datasets from mobile devices. Serving as a dataset for information networks, it is primarily targeted by four types of attacks: DDoS attacks, backdoor attacks, password attacks, and injection attacks.

After analyzing the datasets, the attack data need to be preprocessed and integrated. In this study, we select consecutive 120 h data for experimentation. These data are segmented into 120 groups, with each group representing one hour of attacks, thereby determining the attack frequency and severity. To predict the next network security posture value based on adjacent time values, 118 sets of experimental data can be derived from the 120 groups for prediction experiments.

### 4.2. ER Iterative Algorithm Fusion

Before conducting the experiments, it is necessary to integrate the experimental data to obtain their specific network security posture values. By gradually integrating the indicators as described in Section 3.3, the information within the assessment framework can be consolidated, resulting in the safety state value of the ICS network, thereby providing an understanding of its network security posture. The network security posture values are depicted in Figure 5. After completing the integration process, these data serve as the initial input for the ICS network safety state prediction model, establishing credibility for the next prediction step. The integrated data represent the current network security posture of the system, against which the model’s predictions are compared to assess the prediction accuracy.

### 4.3. Establishment of Industrial Control System Network Security Situation Prediction Model Based on ER and BRB

After obtaining the integrated network security status values, the next step is to establish the prediction model. In this study, two adjacent network security status values are selected as inputs to the model. These status values are then fed into the BRB model to obtain the security prediction values. According to the classification of the basic situation security index of network security released by CNCERT/CC, the results of the industrial control network security prediction model are categorized into five prediction levels: Excellent (A), Good (B), Fair (C), Poor (D), and Critical (E). The transformation of the input data into prediction values is based on the confidence rules of the BRB network security posture prediction model, as described below:(41)$$R_{k}:If\;T\left(k-1\right)\;is{\;B}_{1}^{k}\wedge T\left(k\right)\;is{\;B}_{2}^{k}\;Then\;T\left(k+1\right)\;is\;\left(A,\beta_{I,k}\right),\;\left(B,\beta_{II,k}\right),\;\left(C,\beta_{III,k}\right),\left(D,\beta_{IV,k}\right),\left(E,\beta_{V,k}\right)\;With\;rule\;weight{\;\theta }_{k}\;and\;attribute\;weight{\;\delta }_{1},\delta_{2}$$

In this study, both the rule weights and attribute weights of the model are set to 1, with their initial confidences detailed in Table A1 in Appendix A. The final prediction results obtained from the confidence rules are referenced against the points and values specified in Table 3.

### 4.4. Parameter Optimization Based on Industrial Control System Network Security Situation Prediction Model

The above describes the relevant operations of the experiment. In this study, the fused security situation values obtained in Section 4.2 will be segmented into 118 groups. Subsequently, the security prediction will be conducted using the predictive model outlined in Section 4.3. The first 108 groups of data will serve as training data for the parameter optimization of the prediction model. The remaining 10 groups of data will be used as test inputs to evaluate the prediction accuracy of the model. For the initial parameter optimization, the P-EO algorithm described in Section 3.5 will be employed to optimize the confidence level. The optimized data can be found in Appendix A Table A2.

By comparing the confidence levels before and after optimization, namely, those in Appendix A Table A1 and Table A2, changes in the weights and confidence levels can be observed. These changes occur due to the uncertainty and subjectivity inherent in expert knowledge. Through optimization algorithms, such subjectivity and uncertainty can be reduced.

### 4.5. Comparative Experimentas

After completing the aforementioned steps, validating the BRB prediction model using the P-EO algorithm proposed in this paper involves comparing it with the initial BRB model before optimization. This comparison is depicted in Figure 6.

Figure 6 reveals that the initial BRB model displays significant prediction bias, whereas the BRB model optimized using the P-EO algorithm shows a better alignment with the actual situation. Moreover, it demonstrates an enhanced capability to predict the network security posture and address limitations associated with the expert parameter settings.

To further validate the predictive performance of the proposed method, several other methods are compared. This study selects some typical prediction models, including the Backpropagation Neural Network (BP) based on quantitative data—a mathematical model for distributed parallel information processing [20]. Radial basis function (RBF) is compared, a commonly used machine learning method that utilizes radial basis functions for data processing and nonlinear mapping to perform regression predictions [23]. The random forest (RF) prediction model is compared, which predicts samples by statistically evaluating the predictions of each decision tree and selecting the final prediction result through a voting mechanism [24]. Two optimization algorithms commonly used in the BRB are also considered: BRB-based Whale Optimization Algorithm (WOA) and Population-based Covariance Matrix Adaptation Evolution Strategy (P-CMA-ES) [35,36]. Both algorithms have demonstrated an effective optimization performance in the BRB. The prediction results of each method are shown in Figure 7.

To evaluate the predictive performance of each model regarding the network security posture, the mean squared error (MSE), root mean squared error (RMSE), and mean absolute percentage error (MAPE) between the actual and predicted values are computed. Each model is subjected to 10 rounds of testing to reduce randomness, and their average values are presented in Table 4. From Figure 7 and Table 4, it is evident that the method proposed in this paper achieves a closer proximity to the actual values compared to the other methods. The MSE, RMSE, and MAPE values were superior to those of the other methods.

From Table 4, it is evident that the method proposed in this paper yields favorable predictive results compared to the other methods. Moreover, the operations of our method are interpretable, unlike those of artificial intelligence, which operate as black boxes. The prediction errors of our method are also comparable to those of the other two BRB model optimization algorithms. Therefore, further comparisons are necessary. By dividing the 118 sets of data into training samples comprising 108, 98, and 88 sets, and testing samples comprising 10, 20, and 30 sets, respectively, the predictive performance can be evaluated based on the MSE values. The comparisons are shown in Table 5.

From Table 5, it is evident that the P-EO optimization algorithm shows lower MSE values compared to the two other optimization algorithms, even with fewer training samples. This suggests higher predictive accuracy. These experiments underscore the advantage of the predictive model proposed in this paper in scenarios with limited samples. Moreover, it effectively addresses challenges related to expert uncertainty while enhancing prediction accuracy.

## 5. Conclusions

This paper analyzed the structure of an ICS and its actual network security posture, establishing a four-level evaluation framework to facilitate information integration. Through the iterative process of ER, information within the framework was integrated. By establishing an ICS network security prediction model based on the BRB, this study aimed to reduce the shortcomings of expert knowledge in parameter setting by using the P-EO optimization algorithm to optimize the model. This approach effectively utilizes semi-quantitative and uncertain information, thereby reducing expert uncertainty. The experimental results show that the prediction model proposed in this paper performs better in predicting the ICS safety compared with other methods, especially when there is less historical data. However, to achieve more accurate predictions, additional historical information may be necessary as input to the model, potentially leading to the BRB model combination explosion problem and decreased prediction efficiency. Moreover, during optimization, optimization algorithms may significantly alter expert predictions, which could reduce interpretability. Future research directions include reducing the number of rules by adjusting the BRB rule combinations to address the combination explosion problem and enhancing the interpretability by introducing reasonable conditions to constrain optimization algorithms.

## Figures and Tables

**Figure 1 sensors-24-04716-f001:**
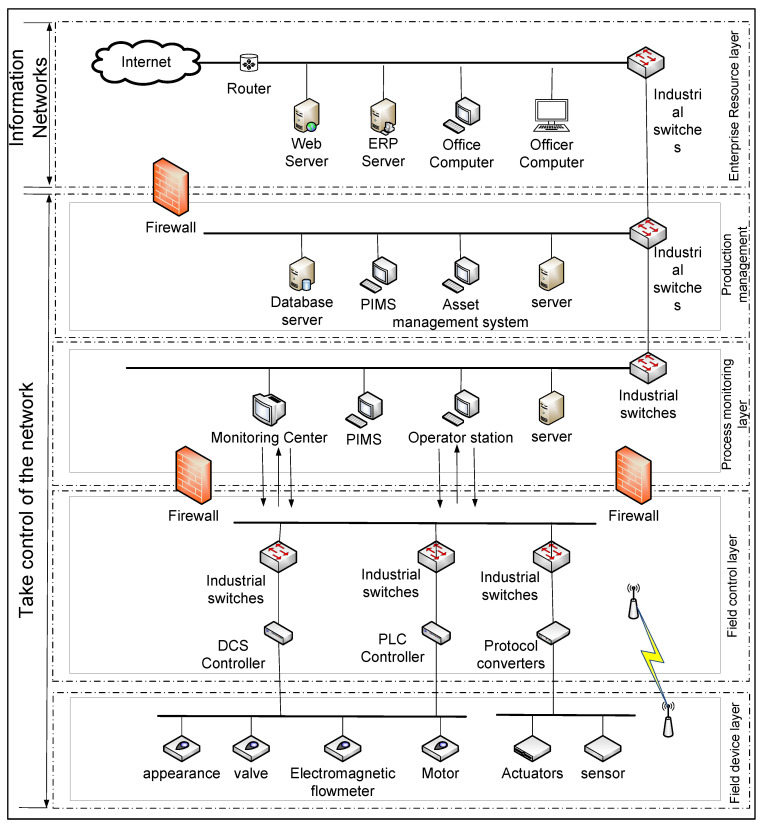
Industrial control system topology diagram.

**Figure 2 sensors-24-04716-f002:**
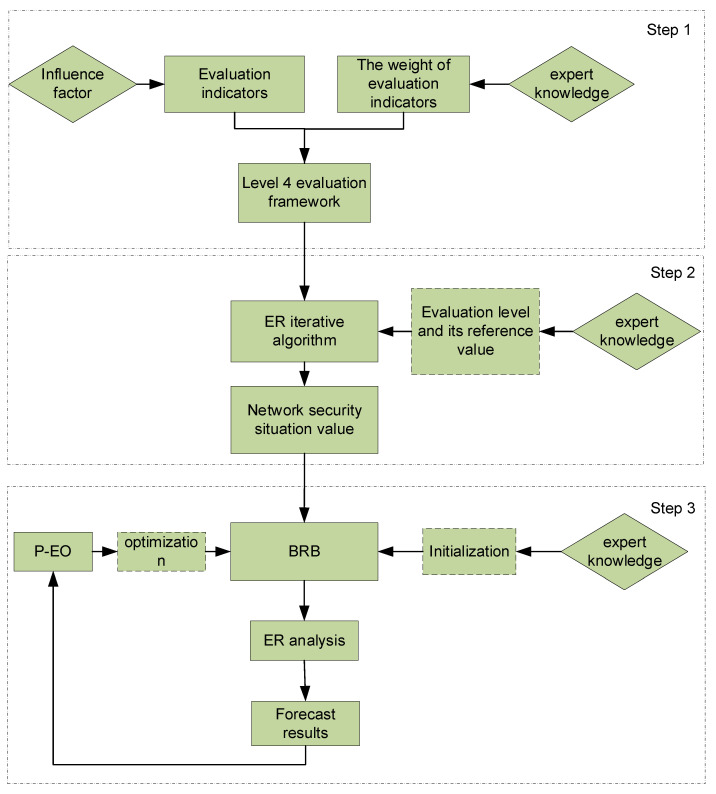
Prediction process of industrial control system network security posture.

**Figure 3 sensors-24-04716-f003:**
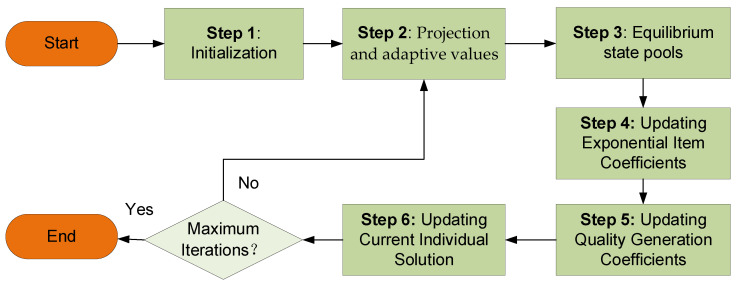
Computational process of the P-EO optimization algorithm.

**Figure 4 sensors-24-04716-f004:**
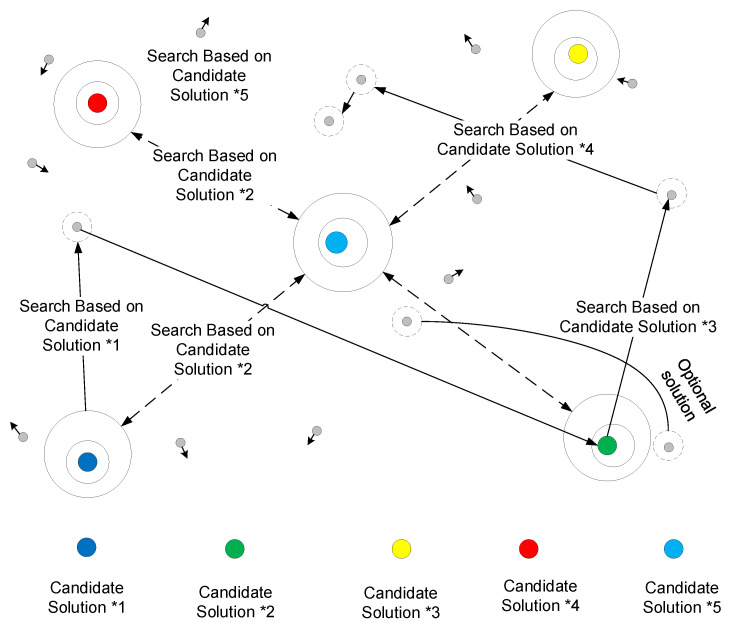
Schematic diagram of the P-EO algorithm selecting the equilibrium state.

**Figure 5 sensors-24-04716-f005:**
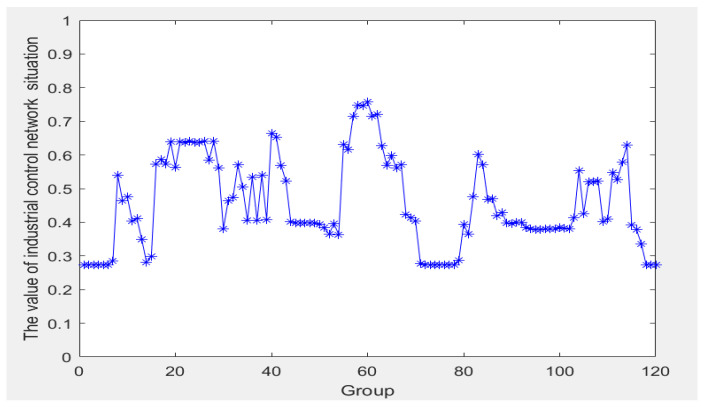
ICS network security situation values.

**Figure 6 sensors-24-04716-f006:**
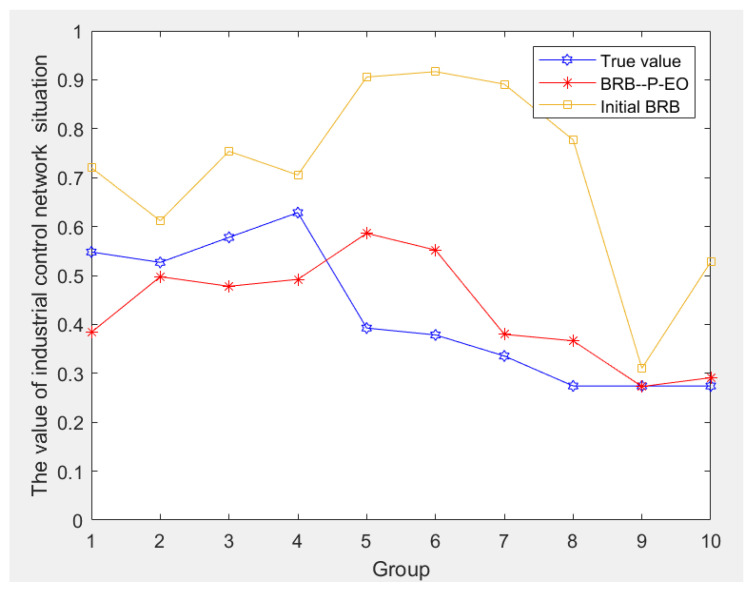
Comparison curve between initial BRB and optimized BRB.

**Figure 7 sensors-24-04716-f007:**
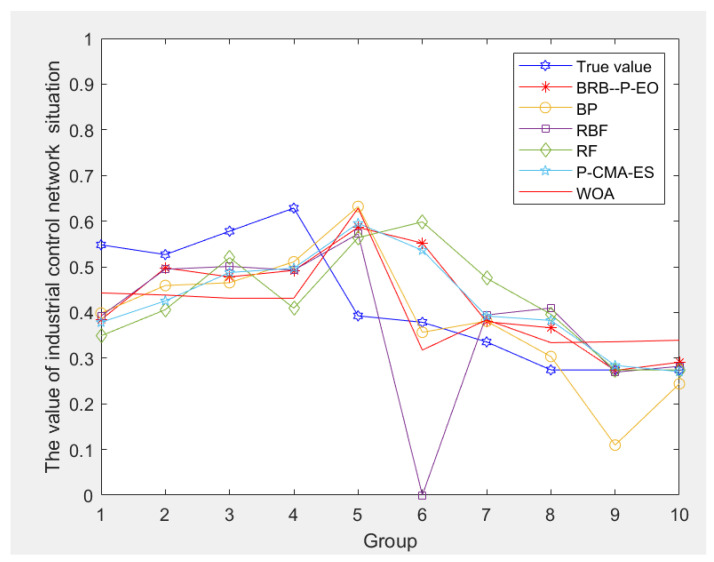
Comparison results of different models’ predictions.

**Table 1 sensors-24-04716-t001:** Parameter table.

Parameter	Meaning
$r_{j}$	The *j*th evaluation indicator
$\alpha_{n,j}$	Confidence level obtained under the *j*th evaluation indicator
Θ	Global ignorance
$D_{n}$	The assessment grade is n
$U_{n,j}$	The *j*th evaluation indicator is the basic probability quality
$U_{\vartheta ,j}$	The basic probability quality that is not assigned to the evaluation scale in the *j*th evaluation indicator
${\bar{U}}_{\vartheta ,j}$	With the exception of the *j*th evaluation indicator, the basic probability quality that is not assigned to the evaluation scale
${\tilde{U}}_{\vartheta ,j}$	The incompleteness of the jth assessment indicator
$U_{n,r\left(2\right)}$	After the fusion of the two evaluation indicators, the basic probability quality of the evaluation grade is *n*
${\tilde{U}}_{\vartheta ,r(2)}$	The joint probability quality assigned to the identification framework after the combination of evaluation indicators *r1* and *r2*
${\bar{U}}_{\vartheta ,r(2)}$	The two evaluation indicators are combined and assigned to the joint probability quality of the identification framework
$r\left(2\right)$	The confidence level after the fusion of evaluation indicators *r1* and *r2*
$u\left(P_{n}\right)$	The evaluation of the utility of level $P_{n}$
$C_{i}^{0}$	Optimization variable after the ith random initialization
$C_{max},C_{min}$	The upper and lower limit vectors representing optimization variables
$C_{eq,I},C_{eq,II},C_{eq,III},C_{eq,IV}$	Representations of the four best solutions found up to the current iteration
$C_{eq,ave}$	The average state of four solutions
$a_{1}$	Weight constant coefficient for global search
*sign*	Symbolic function
r, λ	Representation of random number variables
$G_{CP}$	Generation of rate control parameter vector
$C_{k}^{g+1}$	The parameters of the *g+1* generation
$C_{eq}$	The control of the concentration inside the volume in equilibrium
$C_{0}$	The control of the initial concentration of volume at time t0
*F*	Exponential coefficient
$\lambda_{l}$	Liquidity rate
G	The control of the rate of mass generation within the volume
*V*	Control volume
$\nu n_{e}$	The number of variables constrained
xn	The number of equation constraints
$A_{e}$	Parameter vectors

**Table 2 sensors-24-04716-t002:** Industrial control system network security posture assessment framework.

Evaluation Indicators	Level One	Level Two	Level Three	Level Four
Industrial control network (R)	Information network ($r_{1}$) ($\omega_{1}$)	DDoS attack ($r_{11}$) ($\omega_{11}$)	Attack frequency ($r_{111}$) ($\omega_{111}$)	None
			Severity of the attack ($r_{112}$) ($\omega_{112}$)	None
		Backdoor attack ($r_{12}$) ($\omega_{12}$)	Attack frequency ${(r}_{121}$) ${(\omega }_{121}$)	None
			Severity of the attack ($r_{122}$) ($\omega_{122}$)	None
		Password attack ($r_{13}$) ($\omega_{13}$)	Attack frequency ($r_{131}$) ($\omega_{131}$)	None
			Severity of the attack ($r_{132}$) ($\omega_{132}$)	None
		Injection attack ($r_{14}$) ($\omega_{14}$)	Attack frequency ($r_{141}$) ($\omega_{141}$)	None
			Severity of the attack ($r_{142}$) ($\omega_{142}$)	None
	Control network ($r_{2}$) ($\omega_{2}$)	Historical/real-time Database ($r_{21}$) ($\omega_{21}$)	Vulnerability scanning ($r_{211}$) ($\omega_{211}$)	Attack frequency ($r_{2111}$) ($\omega_{2111}$)
				Severity of the attack ($r_{2112}$) ($\omega_{2112}$)
			Generic scanning ($r_{212}$) ($\omega_{212}$)	Attack frequency ($r_{2121}$) ($\omega_{2121}$)
				Severity of the attack ($r_{2122}$) ($\omega_{2122}$)
			Error data injection ($r_{213}$) ($\omega_{213}$)	Attack frequency ($r_{2131}$) ($\omega_{2131}$)
				Severity of the attack ($r_{2132}$) ($\omega_{2132}$)
		Asset management system ($r_{22}$) ($\omega_{22}$)	Ransomware ($r_{221}$) ($\omega_{221}$)	Attack frequency ($r_{2211}$) ($\omega_{2211}$)
				Severity of the attack ($r_{22212}$) ($\omega_{22212}$)
			Ransom denial of service ($r_{222}$) ($\omega_{222}$)	Attack frequency ($r_{2221}$) ($\omega_{2221}$)
				Severity of the attack ($r_{2222}$) ($\omega_{2222}$)
			Resource discovery ($r_{223}$) ($\omega_{223}$)	Attack frequency ($\;r_{2231}$) ($\omega_{2231}$)
				Severity of the attack ($r_{2232}$) ($\omega_{2232}$)
		Industrial gateway ($r_{23}$) ($\omega_{23}$)	Modbus register read ($r_{231}$) ($\omega_{231}$)	Attack frequency ($r_{2311}$) ($\omega_{2311}$)
				Severity of the attack ($r_{2312}$) ($\omega_{2312}$)
			Brute force attack ($r_{232}$) ($\omega_{232}$)	Attack frequency ($r_{2321}$) ($\omega_{2321}$)
				Severity of the attack ($r_{2322}$) ($\omega_{2322}$)
			Reverse shell attack ($r_{233}$) ($\omega_{233}$)	Attack frequency ($r_{2331}$) ($\omega_{2331}$)
				Severity of the attack ($r_{2332}$) ($\omega_{2332}$)
			Man-in-the-middle attack ($r_{234}$) ($\omega_{234}$)	Attack frequency ($r_{2341}$) ($\omega_{2341}$)
				Severity of the attack ($r_{2342}$) ($\omega_{2342}$)

**Table 3 sensors-24-04716-t003:** Reference points and values of prediction results.

Reference Points	A	B	C	D	E
Values	0.2	0.4	0.6	0.8	1

**Table 4 sensors-24-04716-t004:** Average MSE values of different models.

Model	Initial BRB	BRB-P-EO	BP	RBF	RF	BRB-P-CMA-ES	WOA
MSE	0.1254	0.0135	0.0374	0.0247	0.0218	0.0145	0.0153
RMSE	0.3515	0.1162	0.1294	0.1517	0.1471	0.1201	0.1599
MAPE	79.92%	26.97%	24.74%	35.79%	28.92%	26.17%	24.81%

**Table 5 sensors-24-04716-t005:** Average MSE values of optimization algorithms.

Model	BRB-P-EO	BRB-P-CMA-ES	WOA
10 sets (MSE)	0.0135	0.0145	0.0153
20 sets (MSE)	0.0090	0.0112	0.0121
30 sets (MSE)	0.0083	0.0110	0.0095

## Data Availability

The data presented in this study are available on request from the corresponding author.

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
