# Peer review of "Network Security Prediction of Industrial Control Based on Projection Equalization Optimization Algorithm"

_sensors, 2024, doi:10.3390/s24144716_

Round 1
Reviewer 1 Report
Comments and Suggestions for Authors
1. The text picture font is too small. The font color in Figure 2 is too bright to read clearly; In Figure 3, the arrows of step3-step4 and step4-step5 can be straight arrows, and the arrows can be shortened appropriately, so the overall picture is not compact; The left note in the picture of experimental results can be placed above the picture for a more intuitive view.
2. Wrong format: for example, part (1)(2)(3)... Indented sequence number is inconsistent; The text.... shown in Table 3. ...... The space between Table and 3. is incorrect.
3. The analysis of Figure 7 in this paper is not in-depth enough to reflect the excellence of the proposed method compared with other methods
Comments on the Quality of English Language
Some English expressions are not accurate. It is necessary to seek professional native English speakers for polishing and revision.
Reviewer 2 Report
Comments and Suggestions for Authors
The paper presents a model to predict the security status of Industrial Control Systems (ICSs). Analyzing the paper, I identified the following issues:
1. Clear definitions of the terms “security status” (lines 9-10), “security situation value” (line 12), or “network security posture” (line 35) need to be specified. These terms are confusingly and inconsistently used in the paper.
2. The references are confusingly cited in the body text. For example, instead of “[1]” it is written “1.” (line 30), or instead of “[2]” it is written “2.” (line 33). Please check the entire manuscript.
3. Lines 14, 267: What does “relevant experts” mean? How do they specify the security levels or initial parameters? On what basis?
4. Lines 20-21: it is written “this paper demonstrates certain advantages in predicting small sample data”, but this seems not to be linked with this paper. What exactly does “predicting small sample data” mean in the context?
5. The optimization model (i.e., the objective function(s) and constraints) used by the P-EO is not precisely specified.
6. The P-EO optimization algorithm used to update the initial parameters is not detailed. It has to be supplemented with mathematical proofs.
7. Line 109: What does “single knowledge” mean in the context?
8. Table 1 contains many inconsistencies. Some examples may be seen in line 14 (Ci0) which is described by “Perform random initialization within the upper and lower bounds of optimization variables” or line 19 where “sign” is described as a “symbolic function”. Moreover, what “Liquidity rate” represent? Also, the reason for retaining only four solutions (Ceq I, Ceq II, Ceq III, Ceq IV) is not specified.
9. Which is the timestep used to predict the “security status”? The proposed procedure is done every hour, every day? On what basis this timestep is selected?
10. Model (1) is unclear.
Comments on the Quality of English Language
English needs serious polishing.
Reviewer 3 Report
Comments and Suggestions for Authors
The authors analyzed the security status of ICS, integrated information through the iterative process of ER, and optimized the network security prediction model BRB, which has a certain workload. However, the following issues should be addressed:
1. Reference should be provided for the comparative algorithms mentioned in P-CMA-ES, WOA, etc.
2. By dividing the sets of data into training samples comparing 108, 98, and 99 sets, and testing samples comprising 10, 20, and 30 sets respectively”, The author should explain why when the training sample for the third group is 99 and the testing sample is 30, their sum does not equal the total number of samples 118.
3. Please modify Figure 4 with the legend annotation "Schematic diagram of P-EO algorithm. optimization principle." The legend annotation should not be in the form of multiple sentences.
4. The author proposed the P-EO algorithm but did not provide a detailed explanation of its role. It is inappropriate to simply summarize and explain it as a small step in step 2. The author should emphasize or compare it with the EO algorithm to highlight the role of P in the P-EO algorithm.
5. A P-EO optimization algorithm was proposed in the article to improve the BRB model, but the reference to P-EO was made in the article "There, the paper ends to optimize the predictive model parameters using the P-EO algorithm 37..." Should we use the basic algorithm EO instead of P-EO here? In the schematic diagram of P-EO algorithm in Figure 4, the role of P is not reflected in the figure, but only a restatement of the EO algorithm flowchart in reference 37.
Round 2
Reviewer 1 Report
Comments and Suggestions for Authors
The paper has been revised according to the comments and is ready for publication.
Author Response
Dear reviewer, thank you very much for your review! We are very grateful for your recognition of our research work.
Reviewer 2 Report
Comments and Suggestions for Authors
The authors have successfully solved most of my comments and concerns. However, two issues still need to be solved:
a) In my previous Comments 9, I asked the following question, that was not answered: " On what basis this timestep is selected?". What I mean is: how did you decide to use a timestep of two hours? Why not 20 minutes? Why nor three days?
b) In my previous Comments 4 I asked "What exactly does “predicting small sample data” mean in the context?" because the sequence "predicting small sample data" does not mean what the authors pretend to (the authors responded: " Predicting a small sample of data means that it is safe to predict it when there is little historical data", which is wrong). Please explain.
Comments on the Quality of English Language
English still needs polishing.
